# Extracellular Enzymatic Activities of Oceanic Pelagic Fungal Strains and the Influence of Temperature

**DOI:** 10.3390/jof8060571

**Published:** 2022-05-26

**Authors:** Katherine Salazar Alekseyeva, Gerhard J. Herndl, Federico Baltar

**Affiliations:** 1Department of Functional and Evolutionary Ecology, University of Vienna, 1030 Vienna, Austria; gerhard.hernld@univie.ac.at; 2Department of Marine Microbiology and Biogeochemistry, Royal Netherlands Institute for Sea Research (NIOZ), University of Utrecht, 1790 Texel, The Netherlands

**Keywords:** marine fungi, total extracellular enzymatic activity, kinetics, maximum velocity, half-saturation constant

## Abstract

Although terrestrial and aquatic fungi are well-known decomposers of organic matter, the role of marine fungi remains largely unknown. Recent studies based on omics suggest that marine fungi potentially play a major role in elemental cycles. However, there is very limited information on the diversity of extracellular enzymatic activities performed by pelagic fungi in the ocean and how these might be affected by community composition and/or critical environmental parameters such as temperature. In order to obtain information on the potential metabolic activity of marine fungi, extracellular enzymatic activities (EEA) were investigated. Five marine fungal species belonging to the most abundant pelagic phyla (Ascomycota and Basidiomycota) were grown at 5 °C and 20 °C, and fluorogenic enzymatic assays were performed using six substrate analogues for the hydrolysis of carbohydrates (β-glucosidase, β-xylosidase, and *N*-acetyl-β-D-glucosaminidase), amino acids (leucine aminopeptidase), and of organic phosphorus (alkaline phosphatase) and sulfur compounds (sulfatase). Remarkably, all fungal strains were capable of hydrolyzing all the offered substrates. However, the hydrolysis rate (V_max_) and half-saturation constant (K_m_) varied among the fungal strains depending on the enzyme type. Temperature had a strong impact on the EEAs, resulting in Q_10_ values of up to 6.1 and was species and substrate dependent. The observed impact of temperature on fungal EEA suggests that warming of the global ocean might alter the contribution of pelagic fungi in marine biogeochemical cycles.

## 1. Introduction

Fungi are eukaryotic and osmoheterotrophic organisms depending on organic matter to grow and obtain energy [1]. Osmotrophy involves the secretion of different enzymes to break down complex biological polymers into smaller monomers that can then be taken up through the cell wall [2]. Due to this conversion of organic matter, osmoheterotrophs such as fungi should be major players in the recycling of organic matter. In the marine environment, most of the research on extracellular enzymatic activity (EEA) has been focused on prokaryotes [3,4]. Only a few studies have reported on fungal EEA related to phytoplankton blooms [5] or the degradation of plant-derived matter [6]. As a result, the ecological role of fungi in the biogeochemistry of the oceans, which represents the largest habitat in the Earth’s biosphere, remains poorly known [7]. This contrasts with recent evidence of fungal biomass dominating the bathypelagic marine snow [8]. Additionally, recent studies based on omics suggest that pelagic fungi harbor genes indicative of an active role in marine biogeochemical cycles [9]. It has also been shown that a large variety of carbohydrate active enzymes (CAZymes) are expressed in marine pelagic fungi [10]. Still, there is very limited information on the diversity of EEA in pelagic fungi in the ocean, and how it might be affected by community composition and/or environmental parameters such as temperature.

Approximately 70% of the Earth’s biosphere is composed of persistently cold environments, from the deep sea to polar regions [11,12]. Depending on the optimal growth temperature, organisms living there can be psychrophilic or psychrotrophic [13]. These organisms need to be well adapted to low temperatures, low nutrient availability, and light seasonality [14,15]. Moreover, as low temperatures influence the biochemical reaction rates, organisms must be prepared to overcome those challenges [11].

Here, we investigated the kinetic parameters (V_max_ and K_m_) of the EEA of five oceanic fungal isolates using six different fluorogenic substrate analogues. β-glucosidase and β-xylosidase were used as a proxy for the degradation of plant-derived matter; *N*-acetyl-β-D-glucosaminidase were used for the utilization of animal and fungal chitinous compounds; alkaline phosphatase and leucine aminopeptidase were used for the cleavage of phosphate moieties and peptides, respectively; and sulfatase was used for the degradation of sulfate esters in macromolecules. We used five fungal strains isolated from the oceanic water column, with two strains belonging to the phylum Ascomycota and three to Basidiomycota. We selected these strains because Ascomycota and Basidiomycota are the most abundant pelagic fungal phyla [9,16,17]. Furthermore, the influence of temperature on fungal EEAs was determined.

## 2. Materials and Methods

### 2.1. Culture of Fungi Species

The fungal species *Blastobotrys parvus* (HA 1620), *Metschnikowia australis* (HA 635), *Rhodotorula sphaerocarpa* (HB 738), and *Sakaguchia dacryoidea* (HB 877) were obtained from the Austrian Center of Biological Resources (ACBR). All these species were isolated from Antarctic Ocean waters at temperatures ranging from -1.24 °C to 5.60 °C [18,19,20,21]. *M. australis* and *R. sphaerocarpa* were isolated close to the South Shetland Islands and Marguerite Bay, respectively. The fungus *Rhodotorula mucilaginosa* was isolated from the Atlantic Ocean at 21.03 °C during the Poseidon cruise on board of *RV* Sarmiento de Gamboa in March 2019. The maximum temperature growth reported is 25 °C for *B. parvus* [18] and *M. australis* [19], and 30 °C for *R. mucilaginosa* [22], *R. sphaerocarpa* [21], and *S. dacryoidea* [20]. In order to have fresh cultures, the pure isolates were cultured on yeast malt extract agar [23,24] for one week. Afterwards, an initial amount of each fungus was diluted in artificial seawater (30 g/L sea salts S9883 Sigma-Aldrich, Vienna, Austria) to obtain an OD_660_ ≈ 1 [25]. The optical density (OD) was measured with a UV-1800 Shimadzu spectrophotometer. Then, 10 mL of this fungal culture was inoculated into an autoclaved growth medium containing 2 g/L of glucose, malt extract, peptone, and yeast extract; 35 g/L of artificial sea salts (S9883 Sigma-Aldrich); and 0.50 g/L of chloramphenicol. Afterwards, 150 mL of this medium containing fungi was filled in Schott bottles and to compare the effect of temperature, all strains were grown in triplicate at 5 °C and 20 °C on a rotary shaker (Jeio Tech ISS-7100 Incubated Shaker, Daejeon, South Chungcheong, Republic of Korea). The culture growth was tracked daily by OD. Once the exponential phase was reached, bottles with similar OD values were chosen in triplicate for further analysis (EEA and biomass).

### 2.2. Determining Extracellular Enzymatic Activity and Fungal Biomass

Fluorogenic substrate analogues such as 4-methylumbelliferyl β-D-glucopyranoside (M3633 Sigma-Aldrich), 4-methylumbelliferyl β-D-xylopyranoside (M7008 Sigma-Aldrich), and 4-methylumbelliferyl *N*-acetyl-β-D-glucosaminide (M2133 Sigma-Aldrich) were used to estimate the potential activity of the enzymes β-glucosidase (BGL), β-xylosidase (BXY), and *N*-acetyl-β-D-glucosaminidase (NAG), respectively (Table 1). These enzymes can hydrolyze cellulose [26,27], chitin, and xylan [28,29], respectively, thus mediating carbohydrate degradation by marine fungi. The hydrolysis of 4-methylumbelliferyl phosphate (M8883 Sigma-Aldrich) and N-succinyl-Ala-Ala-Pro-Phe-7-amido-4-methylcoumarin (L2145 Sigma-Aldrich) was used to estimate the potential enzymatic activity of alkaline phosphatase (ALP) and leucine aminopeptidase (LAP), respectively. ALP is indicative of the capability of microbes to acquire inorganic phosphorus from organic molecules, and LAP is involved in the hydrolysis of proteins and peptides [30]. Finally, 4-methylumbelliferyl sulfate potassium salt (M7133 Sigma-Aldrich) was used to determine the activity of sulfatase (SUL) degrading sulfate esters in macromolecules. The hydrolysis of these fluorogenic substrates analogues were standardized to the corresponding fluorophores. The fluorophores methylcoumaryl amide (MCA) (A9891 Sigma-Aldrich) and methylumbelliferone (MUF) (M1381 Sigma-Aldrich) were dissolved in 2-methoxyethanol to obtain a final concentration of 100, 50, 10, and 1 μM, and 2000, 1000, 100, and 50 μM.

Sterile microplates of 96 wells with an F bottom and low protein binding (XT64.1, Carl Roth, Karlsruhe, Baden-Wurtemberg, Germany) were used. The standards were distributed to each biological triplicate to establish a standard calibration curve, and each biological triplicate without any addition was used as a blank to determine the background fluorescence of the medium. Serial dilutions of the fluorogenic substrate were established resulting in 12 final concentrations ranging from 100 to 0.05 μM. The fluorescence was measured with a Tecan Infinite 200 PRO at an excitation wavelength of 365 nm and an emission wavelength of 445 nm. An initial measurement was performed, and then, every hour, a measurement was made over a total period of 3 h. Between measurements, the microplates were incubated in the dark at their respective temperature.

For fungal biomass determination, combusted (450 °C; for 6 h) Whatman GF/F filters (WHA1825047 Sigma-Aldrich, 47 mm filter diameter) were individually wrapped in aluminum foil and weighed. Then, 40 mL of each fungal culture triplicate was gently filtered onto a combusted and weighed filter, and was dried at 80 °C for 3 days. Thereafter, the sample was weighed again to determine the fungal biomass as dry weight.

### 2.3. Determination of the Kinetic Parameters of the Extracellular Enzymatic Activity (EEA)

The increase in fluorescence over time (3 h) was transformed into hydrolysis rate [μmol L^−1^ h^−1^] using the equation obtained from the standard calibration lines of MCA and MUF. The resulting hydrolysis rates were fitted directly with the Michaelis–Menten equation using nonlinear least-squares regression analysis with R software [31]. The enzymatic kinetic parameters maximum velocity (V_max_) and half-saturation constant (K_m_) were calculated. To obtain the biomass-specific activity, the V_max_ was normalized to the fungal biomass obtained from the dry weight. Additionally, the Q_10_ value was calculated to identify the dependence of enzyme activity on temperature [32].

### 2.4. Statistical Analyses

For the kinetic parameters V_max_ and K_m_, a one-way analysis of variance (ANOVA) was performed to determine differences among species and temperature. Tukey’s Honestly Significant Difference (Tukey’s HSD) was used for a multiple and simultaneous comparison between species and to identify significance at the species level. A Student’s T-test was used to test the normal distribution of the data. A principal component analysis (PCA) was performed to analyze the fungal enzymatic activity by species and substrate. For this purpose, the prcomp command of the R software was used. To maximize the sum of the variance of the squared loadings, Varimax rotation was performed.

## 3. Results

Remarkably, all the fungal strains were hydrolyzing all the fluorogenic substrates offered (Figure 1 and Figure 2). The kinetic parameters V_max_ and K_m_ of the EEA varied, however, among the different fungal strains and substrates (Figure 3 and Figure 4). Additionally, the response of EEA to temperature was species and substrate dependent (Figure 5 and Figure 6).

### 3.1. Carbohydrate Active Enzymes

#### 3.1.1. β-Glucosidase (BGL)

*S. dacryoidea* exhibited a significantly higher V_max_ for BGL than the other fungal strains (*t*-test; *p* < 0.001) at 5 °C (6.4 ± 3.2 μmol/g biomass/h) and 20 °C (7.4 ± 3.0 μmol/g biomass/h) (Figure 1A). The other fungal strains exhibited generally low BGL activity, particularly at 5 °C. At 20 °C, the BGL activity was slightly higher than at 5 °C, except for *R. mucilaginosa,* which maintained a low BGL activity at both temperatures. Consequently, V_max_ was a species- and temperature-specific value, with Q_10_ values varying between 1.4 and 2.4, except for *R. mucilaginosa* (Figure 5).

The K_m_ was significantly higher in *S. dacryoidea* than in the other fungal strains (*t*-test; *p* < 0.001) and ranged between 139.6 ± 19.1 μM and 78.6 ± 11.9 μM (Figure 3A). Moreover, the K_m_ of *S. dacryoidea* was significantly higher at 20 °C than at 5 °C (*t*-test; *p* < 0.001). Nonetheless, the other two species of the phylum Basidiomycota, *R. sphaerocarpa* and *R. mucilaginosa*, showed low K_m_ values, especially the latter. Finally, for the Ascomycota species, the K_m_ was highest at 5 °C (*t*-test; *p* = 0.01) with 12.9 ± 3.4 μM for *B. parvus* and 38.7 ± 14.6 for *M. australis*.

#### 3.1.2. β-Xylosidase (BXY)

All fungal strains tested were capable to cleave xylose, however, at low rates (Figure 1B). *B. parvus* exhibited a significantly higher V_max_ for BXY than the other strains (*t*-test; *p* < 0.001). In *B. parvus*, the V_max_ was 0.8 ± 0.3 μmol/g biomass/h at 5 °C and 1.0 ± 0.5 μmol/g biomass/h at 20 °C. In the other strains, the hydrolysis rates were higher at 20 °C than at 5 °C, amounting to 0.5 and 0.1 μmol/g biomass/h, respectively. When all the fungal species were compared, the temperature had a greater effect in *M. australis* and *S. dacryoidea* with Q_10_ values of 2.1 and 1.8, respectively, than in the other fungal strains (Figure 5). For *B. parvus* and both species of the genus *Rhodotorula*, Q_10_ values close to 1 suggested that V_max_ was independent of the temperature.

The K_m_ varied significantly between both temperatures (*t*-test; *p* = 0.90) (Figure 3B). At 5 °C, *S. dacryoidea* showed a high K_m_ (49.7 ± 27.6 μM) followed by *B. parvus* with a K_m_ value of 37.2 ± 8.5 μM. In contrast, at 20 °C, the K_m_ decreased in both *B. parvus* and *S. dacryoidea* but increased in *M. australis*. The other two species, *R. sphaerocarpa* and *R. mucilaginosa*, showed low K_m_ values corresponding to their low V_max_.

#### 3.1.3. *N*-acetyl-β-D-glucosaminidase (NAG)

*B. parvus* exhibited similar NAG hydrolysis rates at 5 °C and 20 °C (*t*-test; *p* = 0.39) (Figure 1C). At 5 °C, the V_max_ was 0.8 ± 0.4 μmol/g biomass/h and at 20 °C 0.7 ± 0.2 μmol/g biomass/h. Although the other strains exhibited low NAG enzymatic activity, it increased from 5 °C to 20 °C, particularly in *M. australis* and *S. dacryoidea*. In *M. australis* the V_max_ increased from 0.1 ± 0.08 μmol/g biomass/h to 0.5 ± 0.2 μmol/g biomass/h and in *S. dacryoidea* from 0.04 ± 0.02 μmol/g biomass/h to 0.1 ± 0.06 μmol/g biomass/h. The Q_10_ values for NAG in *M. australis* and *S. dacryoidea* were 2.1 and 1.1, respectively (Figure 5).

The high NAG enzymatic activity detected in *B. parvus* coincided with a high K_m_ (Figure 3C). At 5 °C, the K_m_ was 2.9 ± 1.8 μM, while at 20 °C it was 12.1 ± 5.6 μM. At 20 °C, the K_m_ of *M. australis* increased to 8.7 ± 4.6 μM. Thus, the K_m_ values of these two Ascomycota species increased with temperature, but they remained low in the three Basidiomycota species.

### 3.2. Extracellular Enzymes Targeting Proteins, Phosphorus, and Sulfur Compounds

#### 3.2.1. Leucine Aminopeptidase (LAP)

Generally, the LAP activity was higher at 20 °C than at 5 °C in all the fungal strains (*t*-test; *p* < 0.001), except in *M. australis* (*t*-test; *p* = 0.46) (Figure 2A). The LAP activity in *R. sphaerocarpa* at 20 °C was 4.3 ± 1.5 μmol/g biomass/h; however, it was only 1.1 ± 0.2 μmol/g biomass/h (*t*-test; *p* < 0.001) at 5 °C resulting in a Q_10_ of 6.1 (Figure 5). All the other fungal strains had Q_10_ values ranging from 1.9 to 1.1. Thus, the LAP activity was the only extracellular enzymatic activity tested that showed a temperature dependency in all the fungal strains (Figure 5).

*B. parvus* and *S. dacryoidea* exhibited significantly higher K_m_ values than the other fungal strains (*t*-test; *p* < 0.001) (Figure 4A). The K_m_ values varied between 195.4 μM and 41.6 μM in *B. parvus* and between 122.3 μM and 42.2 μM in *S. dacryoidea*. While in *B. parvus,* the K_m_ decreased with increasing temperature, and in *S. dacryoidea,* the K_m_ value incremented with increasing temperature.

#### 3.2.2. Alkaline Phosphatase (ALP)

At 5 °C, *M. australis* exhibited a significantly higher V_max_ (8.8 ± 3.7 μmol/g biomass/h) than the other fungal isolates (*t*-test; *p* = 0.03) (Figure 2B). At 20 °C, however, the V_max_ was about 10-fold lower than at 5 °C (0.6 ± 0.2 μmol/g biomass/h). In contrast, in all the other fungal strains, V_max_ slightly increased with temperature (Figure 2B). The Q_10_ values ranging from 2.6 to 1.0 indicated a moderate temperature dependency of ALP in all the fungal strains examined, except in *M. australis* (Figure 5).

The K_m_ was significantly higher in the Ascomycota than in the Basidiomycota strains (*t*-test; *p* = 0.01) (Figure 4B). *M. australis* exhibited a higher K_m_ at 5 °C (101.5 ± 43.5 μM) than at 20 °C, whereas the K_m_ in *B. parvus* was higher at 20 °C (85.7 ± 10.5 μM) than at 5 °C. The other fungal species belonging to the Basidiomycota phylum exhibited K_m_ values ranging from 55.1 μM to 4.1 μM (Figure 4B).

#### 3.2.3. Sulfatase (SUL)

Sulfatase (SUL) activity was generally low compared to the other extracellular enzymatic activities (Figure 2C). Higher V_max_ values were determined for *B. parvus* and *S. dacryoidea* at 5 °C (1.1 and 0.9 μmol/g biomass/h, respectively) (*t*-test; *p* = 0.01) than at 20 °C (0.4 and 0.3 μmol/g biomass/h, respectively). In contrast, for *M. australis*, *R. mucilaginosa*, and *R. sphaerocarpa*, the SUL activity increased with temperature. The Q_10_ value in *M. australis* was 1.9 while all the other fungal strains were <1.0 (Figure 5). With the exception of *R. sphaerocarpa,* all the fungal strains exhibited higher K_m_ values at 5 °C than at 20 °C (*t*-test; *p ≤* 0.002) (Figure 4C).

### 3.3. Relation between Enzyme Kinetic Parameters, Enzyme Types, and Phylogeny

PCA analysis allowed for the comparison of the five studied marine fungal species and the different extracellular enzymatic activities determined. The explained variance of the dataset was 63.9% (Figure 6). The minor angle between *R. mucilaginosa* and *R. sphaerocarpa* belonging to the same genera (*Rhodotorula*) indicated very similar activity levels and extracellular enzyme characteristics. In contrast, the other fungal strains substantially differed in their extracellular enzymatic activity and enzyme characteristics, independent of their taxonomic affiliation.

## 4. Discussion

All the studied marine pelagic fungi species, *B. parvus, M. australis*, *R. mucilaginosa*, *R. sphaerocarpa*, and *S. dacryoidea*, produced extracellular enzymes to degrade substrate analogues of carbohydrates such as cellulose, chitin, and xylan. Additionally, they all produced enzymes to cleave off amino acids, phosphate, and sulfate esters from organic compounds.

### 4.1. Extracellular Enzymatic Activities of Pelagic Fungal Isolates

#### 4.1.1. Influence of Taxonomy/Diversity on the Different EEAs

Since each organism has specific enzymatic capabilities and substrate preferences [33], extracellular enzymatic activities (EEA) can be used as functional traits to investigate functional diversity [34]. Hence, the different EEAs detected in the studied marine fungi can be used to infer their influence on marine ecological processes. Interestingly, we found that the two species belonging to the genus *Rhodotorula* (*R. mucilaginosa* and *R. sphaerocarpa*) exhibited similar kinetic parameters (V_max_ and K_m_) for the majority of extracellular enzymes (Figure 1, Figure 2, Figure 3, Figure 4 and Figure 6). This might indicate some degree of trait conservation among organisms on the genus level, although more species of this genus need to be investigated before a firm conclusion can be drawn. Differences were observed, however, in the EEAs of all the other fungal strains.

Polysaccharides, as the most abundant organic compound class, but also the most complex one, require a wide range of enzymes to degrade them [35,36]. We measured three EEAs responsible for the cleavage of cellulose, xylan, and chitin (Figure 1). In terrestrial environments, the plant cell wall is composed mainly of cellulose and protected by lignin [37]. In marine ecosystems, cellulose is present in the algae cell wall and covered by distinct polymers [38]. Nonetheless, as marine cellulose is more accessible, but less frequent than other substrates, only specific microorganisms are capable of degrading cellulose [39]. Some studies have reported cellulose degradation by marine fungi like *Arthrinium saccharicola* [40] and *Lulworthia floridana* [41]. Other studies have also described cellulose hydrolysis by wider distributed fungal species, for instance, *Aspergillus niger* [42] and *Trichoderma viren**s* [43]. Vaz, et al. [44] showed that 76% of the studied marine fungi exhibit cellulolytic activity. In this case, even though all the marine fungal species used were able to cleave cellulose, *S. dacryoidea* dominated this EEA (Figure 1A). Hudson [45] stated that each fungi species have different capacities to decompose cellulose due to diverse enzymatic machinery. *S. dacryoidea* exhibited a high K_m_ indicating a low affinity to the substrate [31]. This also suggests that, even though the overall enzymatic activity is high, the substrate dissociates easily from the enzyme [46].

Xylan is a polysaccharide formed by residual monosaccharides called xylose [29]. Similar to cellulose, in terrestrial environments, xylan can be found in plants [47], whereas in the ocean, xylan can be present in algae [48,49]. Fungi can degrade xylose via the oxidoreductase pathway [50] as it is a primary carbon source [28]. Even though the general EEA was low (Figure 1B), we can deduce that the studied marine fungi are capable of releasing enzymes related to the hydrolysis of xylose. Raghukumar, et al. [51] identified low xylanase activity rates of fungal coastal strains. Duarte, et al. [52] also reported xylanase activity of Antarctic fungal strains but highlighted a higher activity of Basidiomycota over Ascomycota strains. In this study, we could not identify a clear difference between these two phyla. Nonetheless, the low K_m_ suggests a high affinity of the enzyme to the substrate at low concentrations (Figure 3B). Thus, it seems likely that pelagic marine fungi might use xylose as a carbon source even when present at low concentrations.

Chitin is one of the most abundant naturally occurring polysaccharides, and it is an essential component of the cell wall of fungi, the exoskeleton of arthropods, the radula of mollusks, and the beak of cephalopods [53,54]. Although the overall chitinase activity was low, we can infer that the studied marine fungi are capable of degrading chitin (Figure 1C). The chitin degradation by marine fungi has been reported in species such as *Lecanicillium muscarium* [55] and *Verticillium lecanii* [56]. Interestingly, the number of fungal enzymes involved in the degradation of chitin is related to the fungal chitin content, which varies strongly between fungal species and is dependent on the growth mode [57]. For instance, the hyphae-like fungal cell wall consists of 10 to 20% of chitin [54], whereas yeast-like fungi have a rather low chitin content of 0.5 to 5% [58]. The filamentous fungus *B. parvus* exhibited high chitinase activity (Figure 1C). Moreover, the low K_m_ obtained for this species suggests that only a low substrate concentration is needed to saturate its chitinase. Thus, pelagic marine fungi likely use chitin as a carbon source even when present at only low concentrations.

Leucine aminopeptidase is a critical biological enzyme due to its key role in the degradation of proteins [59]. In the present study, this enzymatic activity was different for each fungal species. Despite the majority of microbial LAP being intracellular, extracellular enzymes have been reported in filamentous fungi [60]. All the species we tested showed LAP activity with *R. sphaerocarpa* exhibiting the highest V_max_. The substrate concentration needed to achieve half V_max_ varied among species (Figure 4A). Although *B. parvus* had one of the lowest V_max_, its K_m_ was the highest (Figure 2A and Figure 4A). In contrast, *R. sphaerocarpa* exhibited high substrate affinity and a low K_m_ (Figure 4A) Thus, as the kinetics for LAP varied among the fungal species, the protein hydrolysis in the ocean might be species dependent.

Inorganic phosphate (Pi) is the preferred phosphorus source for microbial uptake. In surface waters, however, Pi frequently limits phytoplankton productivity [61]. To overcome this P-limitation, microorganisms use dissolved organic phosphorus (DOP) [62]. For prokaryotic microorganisms, Baltar et al. [63] reported that irrespective of the phosphate bioavailability, the activation of alkaline phosphatase was related to sporadic pulses of organic matter. Thus, a high K_m_ might be beneficial to allow for high cleavage rates when organic substrate availability is high. The species *R. sphaerocarpa*, *R. mucilaginosa*, and *S. dacryoidea*, belonging to the phylum Basidiomycota, exhibited a low enzymatic activity but a high K_m_ (Figure 2B and Figure 4B). In contrast, the species of the phylum Ascomycota (*B. parvus* and *M. australis*) seem to be more suitable to overcome this P-limitation, but a higher amount of substrate, and hence of organic matter, might be needed for this purpose.

In living organisms, sulfur is the sixth most abundant element as it can be found in amino acids, such as cysteine and methionine, but also in polysaccharides and proteoglycans [35]. In contrast to terrestrial polysaccharides, several marine polysaccharides, especially in the cell wall of macroalgae, are highly sulfated [35]. The terrestrial fungus *Fusarium proliferatum* was reported to produce sulfatase for the assimilation of sulfated fucoidans of brown algae [64]. In the present study, the species of the genus *Rhodotorula* maintained a low sulfatase activity, whereas it varied among the other species. The V_max_ and K_m_ were high in *S. dacryoidea,* indicating fast hydrolysis and low substrate affinity. As all the fungal species showed different sulfatase activity, we can deduce that these marine fungi might be capable to use sulfated amino acids as well as carbohydrates.

In this study, even though we analyzed just a few hydrolysis possibilities, the functional diversity seems to be broad in marine fungi. As suggested by Berlemont [65] and Baltar et al. [10], marine fungi are potentially involved in carbohydrates’ degradation. Based on our results, we can infer that marine pelagic fungi are actively utilizing carbohydrates such as cellulose, chitin, and xylose, as well as some carbohydrates with sulfur content, potentially of algal origin [47].

The ocean is a complex and diverse ecosystem composed of microorganisms such as archaea, bacteria, fungi, protists, and viruses. As osmoheterotrophic organisms, fungi might provide intermediate decomposition products needed for other microorganisms. These also might lead to the proliferation or inhibition of other microbes as well as enzymatic activities. As marine fungi can utilize a wide range of organic substrates [66], nutrient availability can impact the magnitude and distribution of extracellular enzymatic activities [33].

#### 4.1.2. Temperature Influence

Temperature is considered one of the most important abiotic factors because it influences essentially all biochemical reactions [67]. Enzymes are sensitive to temperature [46] influencing the kinetics along with the substrate binding property and stability [33,68]. According to the Van’t Hoff rule, a temperature increase of 10 °C can double a reaction rate. Q_10_ values lower than 1.0 would indicate a reaction rate completely independent of temperature, whereas values above 1 indicate thermodependency [69].

The results obtained in this study indicate that the majority of enzymatic activities were lower at 5 °C than at 20 °C (Figure 1 and Figure 2). In general terms, at 5 °C the species *R. mucilaginosa* and *R. sphaerocarpa* maintained V_max_ values as low as 0.1 μmol/g biomass/h, whereas the other species exhibited only half of the activity at 5 °C, particularly for BGL and ALP (Figure 1 and Figure 2). This reduced enzymatic synthesis at a low temperature might be due to limited transcriptional and translational activity, limited protein folding, and DNA and RNA secondary structures’ stabilization [11]. *B. parvus*, for all the substrates except ALP, had a higher K_m_ at 5 °C, which suggests that the affinity of this species for some substrates increases with increasing temperature. Taken together, the effect of temperature on the characteristics of extracellular enzymes depends on the fungal species and the type of enzyme.

Psychrophiles microorganisms have evolved a complex range of adaptation strategies, such as production of antifreeze proteins [70] and exopolysaccharides (EPS) [71], high levels of unsaturated fatty acids to maintain the membrane fluidity [72], and certain enzymes adapted to those temperatures [11]. Gerday et al. [73] described a peculiar type of extracellular enzymes known as “cold-adapted enzymes” produced by microorganisms living at low temperatures. For these enzymes, the reaction rate is dependent on the encounter rate of the enzyme and substrate, so it is controlled mainly by diffusion, and it is temperature independent [14,74].

Aghajari et al. [75] suggested that the main structural feature of these “cold-adapted enzymes” is flexibility or plasticity. The structures involved in the catalytic cycle are more flexible, whereas other structures that do not participate in the catalytic cycle might be more rigid [67,76]. For instance, the chitinase of *Glaciozyma antarctica* presented fewer salt bridges and hydrogen bonds, which increased its flexibility [12,77]. Another key structural feature of these enzymes is stability [73,74], with for example, amino acids modifications in key regions of the protein [77,78,79,80]. Nonetheless, there is not a single strategy, as each cold-adapted enzyme can perform different ways to enhance its activity at low temperatures [12,74].

Cold-adapted enzymes have been reported from a wide variety of marine fungi [12,44,52,55,56,77,81,82]. *M. australis* and *R. sphaerocarpa* were one of the few species that showed a noticeable enzymatic activity at 5 °C for ALP and SUL, respectively (Figure 2B,C). *M. australis* is an endemic species of Antarctic waters [19,83], whereas *R. sphaerocarpa* was originally isolated close to Marguerite Bay on the west side of the Antarctic Peninsula but has a wider distribution including the Caribbean Sea [84] and the Andaman Sea [85], among others. Low temperatures exert high selective pressure on endemic organisms [86], such as alkalinity phosphatase in *M. australis*. For this species, the substrate-binding affinity was lower at 5 °C, whereas for *R. sphaerocarpa*, K_m_ was lower at this temperature (Figure 4). This suggested that the enzyme–substrate complex of *M. australis* ALP is more effective at higher substrate concentration typical for Antarctic waters known as a major high-nutrient low-chlorophyll region in the global ocean [87].

Fungal cold-adapted chitinases have been previously reported [77,88,89]. Ramli et al. [77] found that the chitinase sequence of *G. antarctica* had a low sequence identity with other chitinases. Moreover, they found that the enzyme flexibility was due to certain amino acids substitutions in the surface and loop regions. In this study, at 5 °C, we could only identify a higher chitinase activity for *B. parvus*. For the rest of the species, there was a positive enzymatic activity, but higher at 20 °C.

Microorganisms isolated from cold environments can also display kinetic parameters similar to those of their mesophilic counterparts [69,90]. Ito et al. [91] deduced that a high Q_10_ value (>2) is due to a conformational change in proteins and indicates the need for high activation energy. We found that only for LAP, all the examined fungal species expressed Q_10_ values higher than 1. Generally, the temperature where an enzyme can achieve its highest activity does not match the optimal growth temperature of the microorganism that is producing it [78]. Apparently, cold-adapted species, such as *B. parvus*, *M. australis*, *R. sphaerocarpa*, and *S. dacryoidea,* can respond to a temperature rise by increasing enzymatic activity. According to the Arrhenius equation, temperature can influence the activation energy needed to initiate a chemical reaction, and hence, its rate. At a higher temperature, the molecules gain energy to move faster, which also increases the collisions between enzymes and substrates. As a result, elevated extracellular enzymatic activity at increasing temperatures in the surface waters might lead to changes in cleavage and uptake rate of organic matter in oceanic fungi.

## 5. Conclusions

In the present study, we have shown that different marine fungal strains exhibit varying extracellular enzyme characteristics with V_max_ and K_m_ values varying over a range of one order of magnitude. Although the fungal species were isolated from coastal Antarctic waters (*B. parvus*, *M. australis*, *R. sphaerocarpa*, and *S. dacryoidea*), and hence are potentially adapted to low temperatures, they exhibited higher extracellular enzymatic activity at 20 °C than at 5 °C, with some exceptions. While in some fungal strains the K_m_ values for specific extracellular enzymes were higher at low temperatures, for other enzymes they were lower. Additionally, there was considerable species-specific variability in the extracellular enzymatic activity. Taken together, our study indicates that temperature might be one of most important physical factors controlling marine fungal extracellular enzymatic activity. Thus, species composition and temperature determine the role of marine fungi in organic matter cleavage in the global ocean.

## Figures and Tables

**Figure 1 jof-08-00571-f001:**
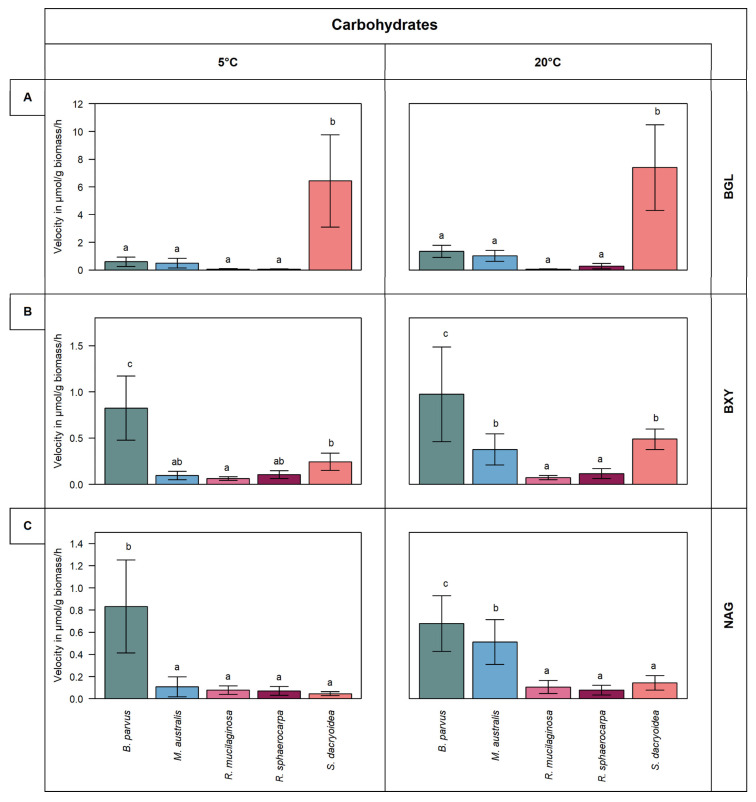
V_max_ in µmol/g biomass/h obtained from the total enzymatic activity for the substrates representing carbohydrates such as (**A**) β-glucosidase (BGL), (**B**) β-xylosidase (BXY), and (**C**) *N*-acetyl-β-D-glucosaminidase (NAG) of five marine fungal isolates *B. parvus*, *M. australis*, *R. mucilaginosa*, *R. sphaerocarpa*, and *S. dacryoidea* at 5 °C and 20 °C. According to Tukey’s HSD, bars denoted by a different letter (a, b, and c) are significantly different (*p* < 0.05), whereas bars denoted by a common letter (ab) are not significantly different.

**Figure 2 jof-08-00571-f002:**
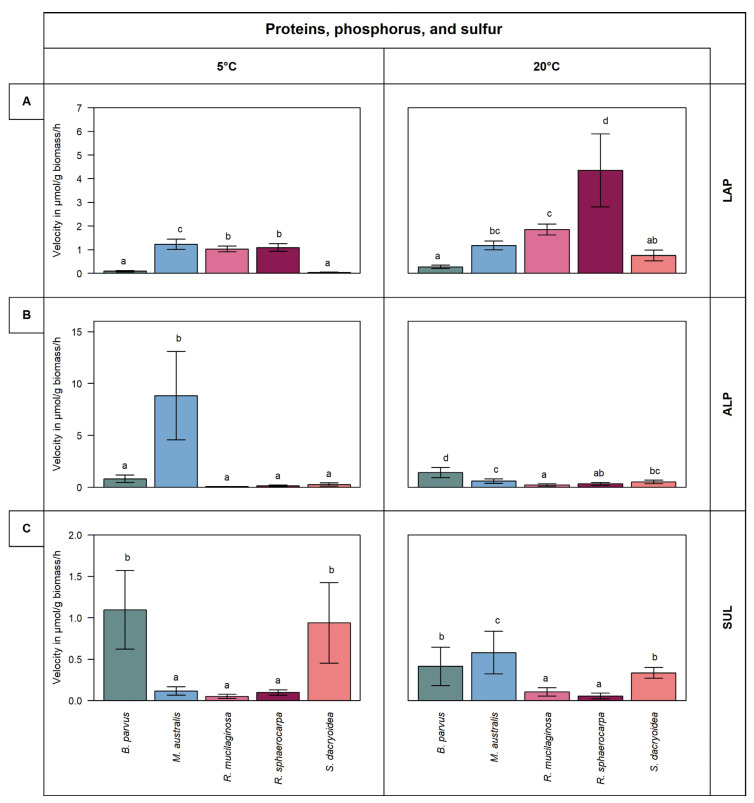
V_max_ in µmol/g biomass/h obtained from the total enzymatic activity for the substrates representing proteins, phosphorus, and sulfur, such as (**A**) leucine aminopeptidase (LAP), (**B**) alkaline phosphatase (ALP), and (**C**) sulfatase (SUL) of the five marine fungal isolates *B. parvus*, *M. australis*, *R. mucilaginosa*, *R. sphaerocarpa*, and *S. dacryoidea* at 5 °C and 20 °C. According to Tukey’s HSD, bars denoted by a different letter (a, b, c, and d) are significantly different (*p* < 0.05), whereas bars denoted by a common letter (ab and bc) are not significantly different.

**Figure 3 jof-08-00571-f003:**
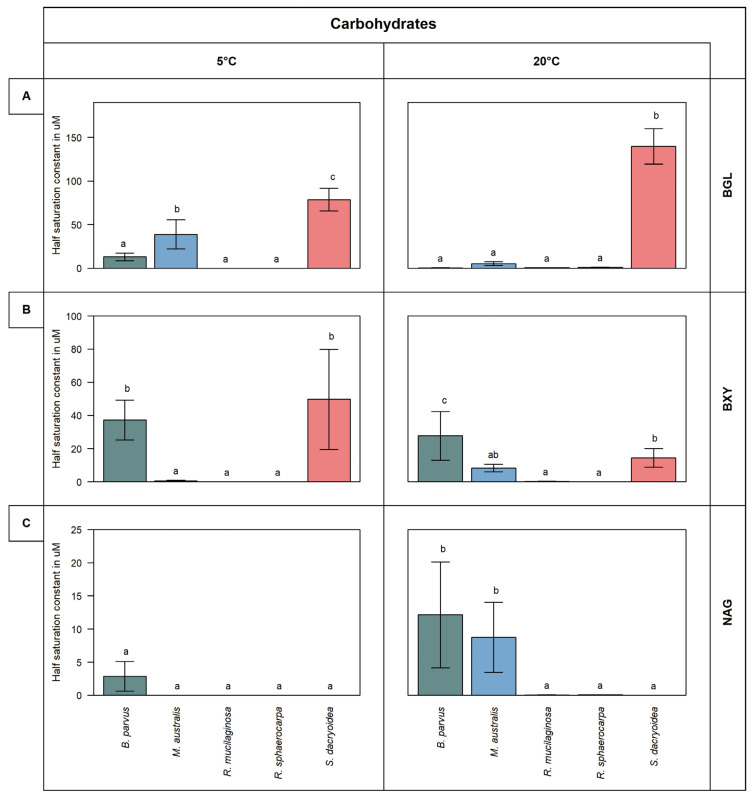
K_m_ in µM obtained from the total enzymatic activity for the substrates representing carbohydrates, such as (**A**) β-glucosidase (BGL), (**B**) β-xylosidase (BXY), and (**C**) *N*-acetyl-β-D-glucosaminidase (NAG) of the five marine fungal isolates (*B. parvus*, *M. australis*, *R. mucilaginosa*, *R. sphaerocarpa*, and *S. dacryoidea*). Measurements were performed at 5 °C and 20 °C in the exponential phase. According to Tukey’s HSD, bars denoted by a different letter (a, b, and c) are significantly different (*p* < 0.05), whereas bars denoted by a common letter (ab) are not significantly different.

**Figure 4 jof-08-00571-f004:**
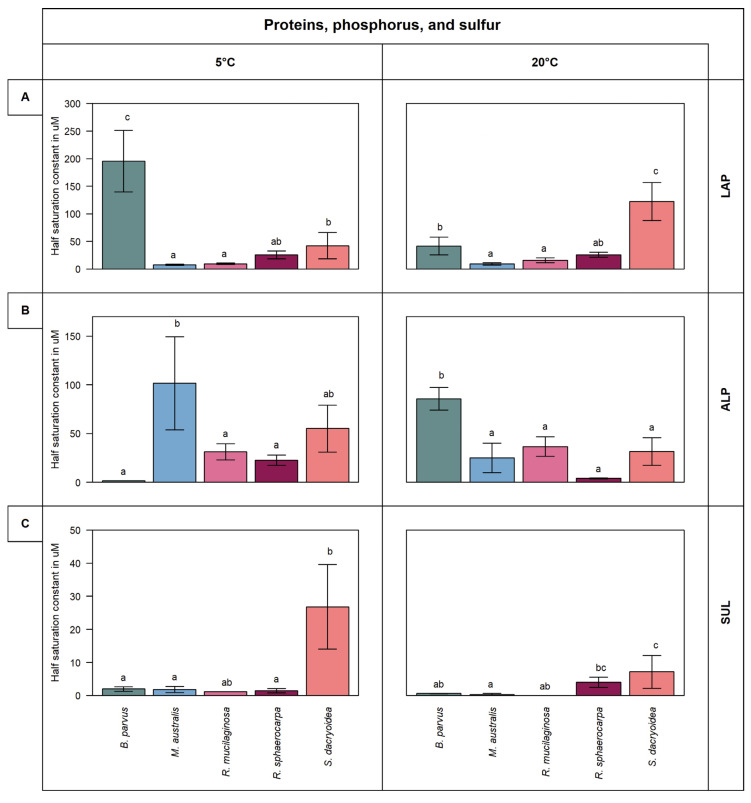
K_m_ in µM obtained from the total enzymatic activity for the substrates representing proteins, phosphorus, and sulfur, such as (**A**) leucine aminopeptidase (LAP), (**B**) alkaline phosphatase (ALP), and (**C**) sulfatase (SUL) of the five marine fungal isolates (*B. parvus*, *M. australis*, *R. mucilaginosa*, *R. sphaerocarpa*, and *S. dacryoidea*). Measurements were performed at 5 °C and 20 °C in the exponential phase. According to Tukey’s HSD, bars denoted by a different letter (a, b, and c) are significantly different (*p* < 0.05), whereas bars denoted by a common letter (ab and bc) are not significantly different.

**Figure 5 jof-08-00571-f005:**
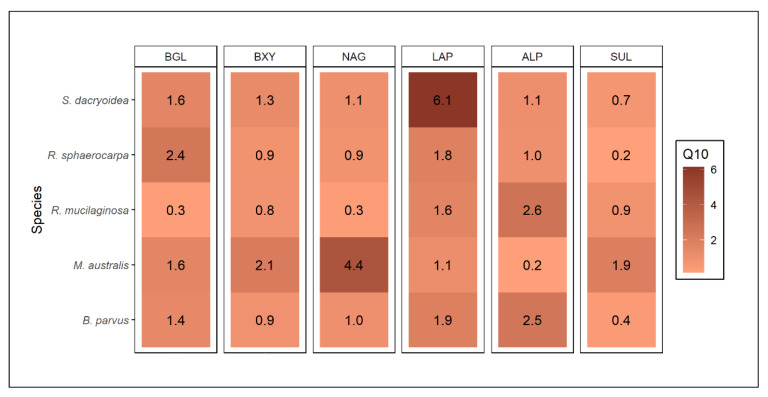
Q_10_ of the normalized total enzymatic activity (V_max_) with the biomass (dry weight) for the substrates β-glucosidase (BGL), β-xylosidase (BXY), *N*-acetyl-β-D-glucosaminidase (NAG), leucine aminopeptidase (LAP), alkaline phosphatase (ALP), and sulfatase (SUL) and the species *B. parvus, M. australis*, *R. mucilaginosa*, *R. sphaerocarpa*, and *S. dacryoidea*.

**Figure 6 jof-08-00571-f006:**
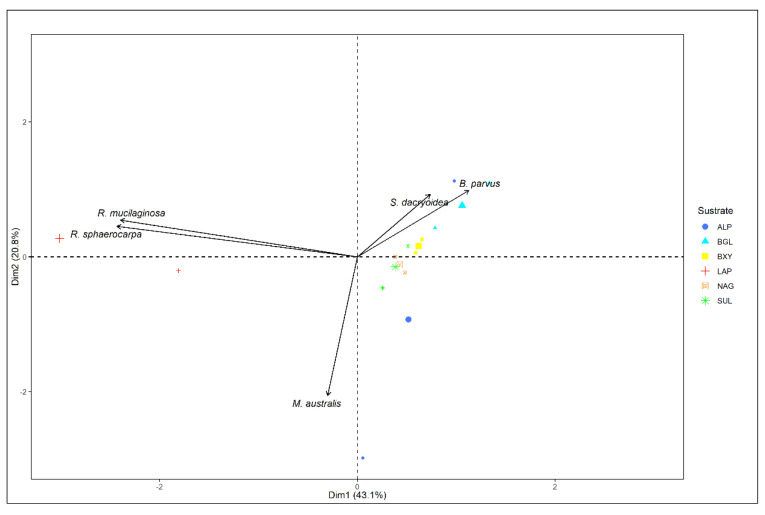
PCA plot from the normalization of the total enzymatic activity (V_max_) with the biomass (dry weight) at 5 °C and 20 °C for all the species. This was compared for each substrate, which corresponds to alkaline phosphatase (ALP), β-glucosidase (BGL), β-xylosidase (BXY), leucine aminopeptidase (LAP), *N*-acetyl-β-D-glucosaminidase (NAG), and sulfatase (SUL).

**Table 1 jof-08-00571-t001:** Targeted enzymes with an analogue fluorogenic substrate and their respective standards, methylumbelliferyl (MUF) and methylcoumaryl (MCA) amide.

Target	Code	Name	Standard
Carbohydrates	BGL	β-glucosidase	MUF
	BXY	β-xylosidase	MUF
	NAG	*N*-acetyl-β-D-glucosaminidase	MUF
Proteins, peptides	LAP	Leucine aminopeptidase	MCA
Phosphorus	ALP	Alkaline phosphatase	MUF
Sulfur	SUL	Sulfatase	MUF

## Data Availability

The raw data supporting the conclusions of this article will be made available by the authors, without undue reservation to any qualified researcher.

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
