# Peer review of "Extracellular Enzymatic Activities of Oceanic Pelagic Fungal Strains and the Influence of Temperature"

_jof, 2022, doi:10.3390/jof8060571_

Round 1
Reviewer 1 Report
The manuscript is overall well written and covers novel aspects. Check the minor typo errors and corrections required. My only comment is to improve the introduction portion, especially add a literature review of some halophilic fungi and thermophilic ones.
Author Response
Response to Reviewer 1 Comments
We acknowledge Reviewer 1 for his/her helpful comments on the paper “Extracellular Enzymatic Activities of Oceanic Pelagic Fungal Strains and the Influence of Temperature”.
The manuscript is overall well written and covers novel aspects.
We appreciate the contribution of Reviewer 1 to this paper.
Check the minor typo errors and corrections required.
We have rechecked the grammar. Mistakes were corrected.
My only comment is to improve the introduction portion, especially add a literature review of some halophilic fungi and thermophilic ones.
In this paper, we would like to focus on the temperature effects as we are preparing another paper where we are talking exclusively about the salinity response. However, following the recommendations of Reviewer 1, we have now improved the introduction and included a paragraph in the manuscript, lines 49-54, it reads: “Approximately 70% of the Earth’s biosphere is composed of persistently cold environments, from the deep sea to polar regions [11, 12]. Depending on the optimal growth temperature, organisms living there can be psychrophilic or psychrotrophic [13]. These organisms need to be well adapted to low temperatures, low nutrient availability, and light seasonality [14, 15]. Moreover, as low temperatures influence the biochemical reaction rates, organisms must be prepared to overcome those challenges [11].” Additionally, we included another paragraph in the discussion, lines 398-401, it reads: “Psychrophiles microorganisms have evolved a complex range of adaptation strategies such as production of antifreeze proteins [70] and exopolysaccharides (EPS) [71]; high levels of unsaturated fatty acids to maintain the membrane fluidity [72], and certain enzymes adapted to those temperatures [11].”
We appreciate the contribution of Reviewer 1 on this paper.

Reviewer 2 Report
Line 69 – what is the temperature for microbial growth. What is the optimum temperature for each fungus?
Line 271-375 – the authors discussed the types of EEAs produced by the marine fungi with the terrestrial fungi counterpart. It is best if the authors could compare the activity between some of the EEAs with other fungal species (i.e Trichoderma virens- https://doi.org/10.1016/j.pep.2018.09.014, Aspergillus niger - https://dx.doi.org/10.1155%2F2014%2F317092)
Line 374-375 – the authors should also note that the temperature also plays an important role in the gene expression, as been described by the previous study – perhaps similar findings for the EEAs genes (https://doi.org/10.1016/j.marenvres.2018.03.007).
Line 368-402 – the authors should also refer/cited some of the papers (https://doi.org/10.3390/jof7070528; https://doi.org/10.3389/fmicb.2016.01408) which have discussed in detail the topic of cold-adapted enzymes.
Line 408-409 – the authors stated that the marine fungi exhibited higher extracellular enzymatic activity at a higher temperature of 20°C than at a lower temperature (5°C), they should discuss why is it happening?
Author Response
Response to Reviewer 2 Comments
We acknowledge Reviewer 2 for his/her helpful comments on the paper “Extracellular Enzymatic Activities of Oceanic Pelagic Fungal Strains and the Influence of Temperature”.
Point 1: Line 69 – what is the temperature for microbial growth. What is the optimum temperature for each fungus?
The optimum temperature of the individual species was not investigated. We added the following explanation in the discussion, lines 437-441: “Generally, the temperature where an enzyme can achieve its highest activity does not match the optimal growth temperature of the microorganism that is producing it [76]. Apparently cold adapted species such as B. parvus, M. australis, R. sphaerocarpa, and S. dacryoidea can respond to a temperature rise by increasing enzymatic activity.” However, we included information about the maximum temperature growth in the methods, lines 75-76, it reads: “The maximum temperature growth reported is 25°C for B. parvus [18] and M. australis [19], and 30°C for R. mucilaginosa [22], R. sphaerocarpa [21], and S. dacryoidea [20].”
Point 2: Line 271-375 – the authors discussed the types of EEAs produced by the marine fungi with the terrestrial fungi counterpart. It is best if the authors could compare the activity between some of the EEAs with other fungal species (i.e Trichoderma virens- https://doi.org/10.1016/j.pep.2018.09.014, Aspergillus niger - https://dx.doi.org/10.1155%2F2014%2F317092)
We have now discussed more EEAs produced by marine fungi, it reads: lines 298-302 “Some studies have reported cellulose degradation by marine fungi like Arthrinium saccharicola [40] and Lulworthia floridana [41]. Other studies have also described cellulose hydrolysis by wider distributed fungal species, for instance, Aspergillus niger [42] and Trichoderma virens [43]. Vaz, et al. [44] showed that 76% of the studied marine fungi exhibit cellulolytic activity.”; lines 313-317 “Raghukumar, et al. [51] identified low xylanase activity rates of fungal coastal strains. Duarte, et al. [52] also reported xylanase activity of Antarctic fungal strains, but highlighted a higher activity of Basidiomycota over Ascomycota strains. In this study, we could not identify a clear difference between these two phyla.”; lines 324-325 “The chitin degradation by marine fungi has been reported in species such as Lecanicillium muscarium [55] and Verticillium lecanii [56].”; and lines 415-416 “Cold adapted enzymes have been reported from a wide variety of marine fungi [12, 44, 52, 55, 56, 77, 81].”
The suggested papers were added, together with:
- Yusof, N.A., N.H.F. Hashim, and I. Bharudin, Cold adaptation strategies and the potential of psychrophilic enzymes from the antarctic yeast, Glaciozyma antarctica PI12. Journal of Fungi, 2021. 7(7): p. 528.
- Hong, J.-H., et al., Investigation of Marine-Derived Fungal Diversity and Their Exploitable Biological Activities. Marine Drugs, 2015. 13(7): p. 4137-4155.
- Meyers, S. and E. Scott, Cellulose degradation by Lulworthia floridana and other lignicolous marine fungi. Marine Biology, 1968. 2(1): p. 41-46.
- Vaz, A.B.M., et al., The diversity, extracellular enzymatic activities and photoprotective compounds of yeasts isolated in Antarctica. Brazilian journal of microbiology : [publication of the Brazilian Society for Microbiology], 2011. 42(3): p. 937-947
- Raghukumar, C., et al., Laccase and other lignocellulose modifying enzymes of marine fungi isolated from the coast of India. 1994.
- Duarte, A., et al., Taxonomic assessment and enzymes production by yeasts isolated from marine and terrestrial Antarctic samples. Extremophiles, 2013. 17(6): p. 1023-1035.
- Fenice, M., The psychrotolerant Antarctic fungus Lecanicillium muscarium CCFEE 5003: A powerful producer of cold-tolerant chitinolytic enzymes. Molecules, 2016. 21(4): p. 447.
- Fenice, M., et al., Chitinolytic activity at low temperature of an Antarctic strain (A3) of Verticillium lecanii. Research in Mi-crobiology, 1998. 149(4): p. 289-300.
- Ramli, A.N.M., et al., Structural prediction of a novel chitinase from the psychrophilic Glaciozyma antarctica PI12 and an analysis of its structural properties and function. Journal of computer-aided molecular design, 2012. 26(8): p. 947-961.
- Duarte, A.W.F., et al., Production of cold-adapted enzymes by filamentous fungi from King George Island, Antarctica. Polar Biology, 2018. 41(12): p. 2511-2521.
Point 3: Line 374-375 – the authors should also note that the temperature also plays an important role in the gene expression, as been described by the previous study – perhaps similar findings for the EEAs genes (https://doi.org/10.1016/j.marenvres.2018.03.007).
We have now discussed the importance of temperature in the role of gene expression of EEAs, it reads: lines 391-393 “This reduced enzymatic synthesis at a low temperature might be due to limited transcriptional and translational activity, limited protein folding, and DNA and RNA secondary structures stabilization [11].”; and lines 427-432 “Fungal cold adapted chitinases have been previously reported [77, 87, 88]. Ramli, et al. [77] found that the chitinase sequence of G. antarctica had a low sequence identity with other chitinases. Moreover, they found that the enzyme flexibility was due to certain amino acids substitutions in the surface and loop regions. In this study, at 5°C, we could only identify a higher chitinase activity for B. parvus. For the rest of species, there was a positive enzymatic activity, but higher at 20°C.”;
The suggested paper was read, and similar papers regarding EAAs genes were added:
- D'Amico, S., et al., Psychrophilic microorganisms: challenges for life. EMBO reports, 2006. 7(4): p. 385-389.
- Ramli, A.N.M., et al., Structural prediction of a novel chitinase from the psychrophilic Glaciozyma antarctica PI12 and an analysis of its structural properties and function. Journal of computer-aided molecular design, 2012. 26(8): p. 947-961.
- Plíhal, O., et al., Large propeptides of fungal β-N-acetylhexosaminidases are novel enzyme regulators that must be intracel-lularly processed to control activity, dimerization, and secretion into the extracellular environment. Biochemistry, 2007. 46(10): p. 2719-2734.
- Reyes, F., et al., β-N-Acetylglucosaminidase from Aspergillus nidulans which degrades chitin oligomers during autolysis. FEMS microbiology letters, 1989. 65(1-2): p. 83-87.
Point 4: Line 368-402 – the authors should also refer/cited some of the papers (https://doi.org/10.3390/jof7070528; https://doi.org/10.3389/fmicb.2016.01408) which have discussed in detail the topic of cold-adapted enzymes.
The suggested papers were added, it reads: lines 49-50 “Approximately 70% of the Earth’s biosphere is composed of persistently cold environments, from the deep sea to polar regions [11, 12].”; lines 409-414 “For instance, the chitinase of Glaciozyma antarctica presented fewer salt bridges and hydrogen bonds which increased its flexibility [12, 77]. Another key structural feature of these enzymes is stability [73, 74], with for example, amino acids modifications in key regions of the protein [77-80]. Nonetheless, there is not a single strategy as each cold-adapted enzyme can perform different ways to enhance its activity at low temperatures [12, 74].”; and lines 433-434 “Microorganisms isolated from cold environments can also display kinetic parameters similar to those of their mesophilic counterparts [69, 89]”
Additionally, similar papers regarding cold adapted enzymes were added:
- D'Amico, S., et al., Psychrophilic microorganisms: challenges for life. EMBO reports, 2006. 7(4): p. 385-389.
- Collins, T., et al., Cold‐Adapted Enzymes. Physiology and biochemistry of extremophiles, 2007: p. 165-179.
- Gerday, C., et al., Cold-adapted enzymes: from fundamentals to biotechnology. Trends in biotechnology, 2000. 18(3): p. 103-107.
- Feller, G. and C. Gerday, Psychrophilic enzymes: molecular basis of cold adaptation. Cellular and Molecular Life Sciences CMLS, 1997. 53(10): p. 830-841.
- Ramli, A.N.M., et al., Structural prediction of a novel chitinase from the psychrophilic Glaciozyma antarctica PI12 and an analysis of its structural properties and function. Journal of computer-aided molecular design, 2012. 26(8): p. 947-961.
- Baeza, M., et al., Identification of stress-related genes and a comparative analysis of the amino acid compositions of translated coding sequences based on draft genome sequences of Antarctic yeasts. Frontiers in microbiology, 2021. 12: p. 133.
- DasSarma, S., et al., Amino acid substitutions in cold-adapted proteins from Halorubrum lacusprofundi, an extremely halo-philic microbe from Antarctica. PLoS One, 2013. 8(3): p. e58587.
- Michetti, D., et al., A comparative study of cold-and warm-adapted Endonucleases A using sequence analyses and molecular dynamics simulations. PloS one, 2017. 12(2): p. e0169586.
Point 5: Line 408-409 – the authors stated that the marine fungi exhibited higher extracellular enzymatic activity at a higher temperature of 20°C than at a lower temperature (5°C), they should discuss why is it happening?
We have now explained the general higher enzymatic activity at 20°C compared to 5°C. We added the following explanation: lines 441-444 “According to the Arrhenius equation, temperature can influence the activation energy needed to initiate a chemical reaction, hence, its rate. At a higher temperature, the molecules gain energy to move faster which also increases the collisions between enzymes and substrates.”
We appreciate the contribution of Reviewer 2 on this paper.

Round 2
Reviewer 2 Report
Line 230 - missing some information (Figure 1 and ....)
Line 242 - reference missing.
Line 385 - reference missing.
Line 407 - reference missing.
Line 465 - reference missing.
Line 506 - reference missing.
Line 660 - reference missing.
Line 667 - reference missing.
Line 668 - reference missing.
Line 690 - reference missing.
Line 920 - reference missing.
Author Response
Response to Reviewer 2 Comments
We acknowledge Reviewer 2 for his/her helpful comments on the paper “Extracellular Enzymatic Activities of Oceanic Pelagic Fungal Strains and the Influence of Temperature”.
Line 230 - missing some information (Figure 1 and ....)
The corresponding figure was added (Figure 5).
Line 242 - reference missing.
The corresponding figure was added (Figure 2).
Line 385 - reference missing.
The corresponding figures were added (Figure 1 and Figure 2).
Line 407 - reference missing.
References were present:
“Another key structural feature of these enzymes is stability [73, 74], with for example, amino acids modifications in key regions of the protein [77-80]”
- Gerday, C., et al., Cold-adapted enzymes: from fundamentals to biotechnology. Trends in biotechnology, 2000. 18(3): p. 103-107.
- Feller, G. and C. Gerday, Psychrophilic enzymes: molecular basis of cold adaptation. Cellular and Molecular Life Sciences CMLS, 1997. 53(10): p. 830-841.
- Ramli, A.N.M., et al., Structural prediction of a novel chitinase from the psychrophilic Glaciozyma antarctica PI12 and an analysis of its structural properties and function. Journal of computer-aided molecular design, 2012. 26(8): p. 947-961.
- Baeza, M., et al., Identification of stress-related genes and a comparative analysis of the amino acid compositions of translated coding sequences based on draft genome sequences of Antarctic yeasts. Frontiers in microbiology, 2021. 12: p. 133.
- DasSarma, S., et al., Amino acid substitutions in cold-adapted proteins from Halorubrum lacusprofundi, an extremely halo-philic microbe from Antarctica. PLoS One, 2013. 8(3): p. e58587.
- Michetti, D., et al., A comparative study of cold-and warm-adapted Endonucleases A using sequence analyses and molecular dynamics simulations. PloS one, 2017. 12(2): p. e0169586.
Line 465 - reference missing.
References are present.
Line 506 - reference missing.
Reference 16 is present in line 62
- Amend, A., et al., Fungi in the Marine Environment: Open Questions and Unsolved Problems. mBio, 2019. 10(2).
Line 660 - reference missing.
Line does not exist.
Line 667 - reference missing.
Line does not exist.
Line 668 - reference missing.
Line does not exist.
Line 690 - reference missing.
Line does not exist.
Line 920 - reference missing.
Line does not exist.
We appreciate the contribution of Reviewer 2 on this paper.
